# Efficacy and Safety of a Personalized Vitamin D_3_ Loading Dose Followed by Daily 2000 IU in Colorectal Cancer Patients with Vitamin D Insufficiency: Interim Analysis of a Randomized Controlled Trial

**DOI:** 10.3390/nu14214546

**Published:** 2022-10-28

**Authors:** Sabine Kuznia, David Czock, Annette Kopp-Schneider, Reiner Caspari, Harald Fischer, Dana Clarissa Laetsch, Marija Slavic, Hermann Brenner, Ben Schöttker

**Affiliations:** 1Division of Clinical Epidemiology and Aging Research, German Cancer Research Center (DKFZ), 69120 Heidelberg, Germany; 2Medical Faculty Heidelberg, Heidelberg University, 69117 Heidelberg, Germany; 3Department of Clinical Pharmacology and Pharmacoepidemiology, University Hospital Heidelberg, 69120 Heidelberg, Germany; 4Division of Biostatistics, German Cancer Research Center (DKFZ), 69120 Heidelberg, Germany; 5Clinic Niederrhein (A Clinic of the German Pension Insurance Rhineland), 53474 Bad Neuenahr-Ahrweiler, Germany; 6Clinic Rosenberg (A Clinic of the German Pension Insurance Westfalen), 33014 Bad Driburg, Germany; 7Division of Preventive Oncology, German Cancer Research Center (DKFZ), 69120 Heidelberg, Germany; 8National Center for Tumor Diseases (NCT), German Cancer Consortium (DKTK), German Cancer Research Center (DKFZ), 69120 Heidelberg, Germany

**Keywords:** vitamin D, colorectal cancer, randomized controlled trial, personalized medicine, loading dose, bolus, treatment regimen, calcium, efficacy, safety

## Abstract

A personalized vitamin D_3_ loading dose has not yet been tested in cancer patients. This interim analysis of the randomized, placebo-controlled VICTORIA trial analyzed the first recruited 74 German adults with nonmetastatic colorectal cancer, a tumor surgery within the past year, and 25-hydroxyvitamin D levels (25(OH)D) < 50 nmol/L. Study participants received a loading dose tailored for a baseline 25(OH)D level and BMI in the first 11 days, followed by a maintenance dose of 2000 IU of vitamin D_3_ daily until end of trial week 12. The mean 25(OH)D levels were 27.6, 31.0, and 34.1 nmol/L in the placebo group and 25.9, 63.1, and 75.5 nmol/L in the verum group during screening, visit 1 (end of loading dose), and visit 2 (end of maintenance dose), respectively. The prevalence of 25(OH)D) ≥ 50 nmol/L at visits 1 and 2 was 3.5% and 17.4% in the placebo group and 80.0% and 100% in the verum group. No events of 25(OH)D > 150 nmol/L or hypercalcemia were observed. Hypercalciuria events at visit 1 (*n* = 5 in verum and *n* = 1 in the placebo group; *p* = 0.209) receded after discontinuation of the study medication. The personalized loading dose effectively and safely increased the 25(OH)D levels, and 2000 IU of vitamin D_3_ daily sustained the achieved levels.

## 1. Introduction

Colorectal cancer (CRC) remains a major public health challenge and accounts for more than 60,000 new cases and more than 24,000 deaths per year in Germany [1].

The 25-hydroxyvitamin D (25(OH)D) level is considered the best-established biomarker to determine vitamin D deficiency and insufficiency, which are defined by the US American Institute of Medicine as 25(OH)D levels below 30 nmol/L and below 50 nmol/L, respectively [2]. Vitamin D insufficiency is very common among CRC patients at all stages, not only shortly after cancer treatment but also at least in the first 2 years after surgery [3,4]. Low 25(OH)D levels were found to be strongly associated with a poorer overall survival of CRC patients in a systematic review of cohort studies from the year 2018 and in more recently published cohort studies [5,6,7]. Furthermore, a recently published systematic review and meta-analysis of randomized controlled trials (RCT) showed that vitamin D supplementation significantly improved the progression-free survival of CRC patients (hazard ratio (HR), 95% confidence interval (95% CI): 0.65 (0.36–0.94) [8]. In addition, there are preliminary data suggesting that vitamin D supplementation might increase the efficacy of chemotherapy and alleviate its adverse reactions [9]. As vitamin D insufficiency is usually neither diagnosed nor treated in CRC patients, there is now evidence that it should be treated [5,6,7,8], but the optimal dosing regimen is unknown.

A pragmatic approach is to increase the 25(OH)D levels quickly with a loading dose followed by a maintenance dose. Several clinical trials highlighted that 25(OH)D levels achieved by vitamin D_3_ loading doses strongly depend on the baseline 25(OH)D level and the person’s body weight [10,11,12]. To consider the patient’s body weight is important, because 25(OH)D is stored in adipocyte fat globules to a large extent (100–300 nmol/kg body weight) and, therefore, is of limited availability in the circulatory system of obese patients [13]. However, the literature on trials testing personalized vitamin D_3_ loading doses is sparse, and none has previously been conducted with CRC patients.

We used a personalized vitamin D_3_ loading dose followed by a maintenance dose of 2000 IU per day for 12 weeks in a placebo-controlled RCT enrolling CRC patients with initial vitamin D insufficiency (25(OH)D < 50 nmol/L) and present how effectively the 25(OH)D levels were raised. Furthermore, data on predefined safety outcomes related to serum 25(OH)D levels, serum and urinary calcium levels, and renal function are presented.

## 2. Materials and Methods

### 2.1. Study Design and Participants

In this article, we provide an interim analysis of the ongoing “*Personalized vitamin D supplementation for reducing or preventing fatigue and enhancing quality of life of patients with colorectal tumor-randomized intervention trial*” (VICTORIA, EudraCT-No: 2019-000502-30; DRKS00019907) covering selected secondary trial outcomes related to the efficacy and safety of a personalized vitamin D_3_ intervention. The primary outcome of the VICTORIA trial “cancer-related fatigue” or secondary outcomes related to the quality of life, diseases, or symptoms will not be addressed in this interim analysis.

Details of the study design, including the main and interim analyses, have been reported in the trial protocol [14]. VICTORIA is an ongoing parallel-group, randomized, double-blind, placebo-controlled clinical trial. Overall, 456 colorectal cancer patients aged 18 years and older were recruited from 7 German rehabilitation clinics. The interim analysis included the first 74 enrolled study participants who were recruited between September 2020 and December 2021 in the first three initiated rehabilitation clinics, which are located in the towns Bad Neuenahr-Ahrweiler, Bad Driburg, and Thyrnau (Germany).

Eligible patients had a diagnosis of nonmetastatic colorectal cancer (not stage IV), a tumor surgery within the past year (type of surgery not specified), and vitamin D insufficiency (25(OH)D) levels < 50 nmol/L) at the time of screening. Most of the patients in the interim analysis were recruited before a protocol amendment made for the 25(OH)D level inclusion cut-off and needed even lower, season-standardized 25(OH)D levels < 50 nmol/L (see study protocols for details [14]). Exclusion criteria comprised mainly an already existing supplementation with high-dose vitamin D, high-dose calcium therapy, and medical conditions/concurrent medication contraindicated for vitamin D_3_ therapy according to the Summary of Product Characteristics (see Appendix A).

### 2.2. Intervention

Participants were randomly assigned in a 1:1 ratio to the vitamin D or placebo group. The placebo capsules had the same appearance and almost the same weight. During the first 11 days, a personalized loading dose based on the 25(OH)D level and body mass index (BMI) at screening was administered, followed by a daily maintenance dose of 2000 IU (2 tablets of Dekristol^®^ 1000 IU merged in 1 capsule) until the end of the trial after 12 weeks.

The personalized loading dose was calculated with the equation of Jansen et al., which targets a 25(OH)D level of 80 nmol/L [12], which is consistent with the Endocrine Society’s consensus for the optimal 25(OH)D levels of 75–100 nmol/L [15].
Loading dose = 165 × BMI [kg/m^2^] × (70–baseline 25(OH)D level [nmol/L])

To avoid nonphysiologically high doses of vitamin D_3_ supplements, the loading dose was administered over 11 days in units of 20,000 or 40,000 IU per day (i.e., 1 or 2 capsules of Dekristol^®^ 20,000 IU or placebo) instead of one large bolus. This was primarily justified by emerging findings on vitamin D metabolism and only secondarily by safety concerns [13].

The randomization list was computer-generated and managed by the pharmacy of the Heidelberg University Hospital. Patients and study staff were masked to the group assignment (double-blind trial).

### 2.3. Study Visits

Blood and urine samples were collected at screening (to determine the laboratory test-based in- and exclusion criteria), visit 1 (trial days 12–21, i.e., end of loading dose and end of rehabilitation clinic stay), and visit 2 (trial weeks 13–16, i.e., end of maintenance dose and end of trial).

### 2.4. Outcomes

The following efficacy outcomes were addressed in the interim analysis:Mean difference in the serum 25(OH)D levels between the intervention and placebo groups at visits 1 and 2.Mean difference in the change of the serum 25(OH)D levels from screening to visits 1 and 2.Difference in prevalence of subjects with adequate serum 25(OH)D levels ≥ 50 nmol/L [2] in the intervention group and placebo group at visits 1 and 2.

The following safety outcomes were addressed in the interim analysis:
Difference in frequency of the hypervitaminosis D (25(OH)D levels > 150 nmol/L [2]), hypercalcemia (albumin-corrected serum calcium > 2.65 mmol/L [16]), hypercalciuria (random urine calcium ≥ 0.79 mmol/mmol creatinine [17]), and renal dysfunction (eGFR < 30 mL/min/1.73 m^2^ calculated with the Chronic Kidney Disease Epidemiology Collaboration (CKD-EPI) equation [18]) between the intervention and placebo group at visits 1 and 2.Mean differences in the levels of albumin-corrected serum calcium, urine calcium/creatinine ratio, and eGFR between the intervention and placebo groups at visits 1 and 2.

### 2.5. Laboratory Methods for 25(OH)D Measurements

The biomarker measurements took place in the laboratories the recruiting rehabilitation clinics collaborate with in clinical routine (MVZ Labor Dr Quade & Kollegen GmbH, Cologne; MVZ Labor Passau, Passau; LADR GmbH MVZ, Paderborn). All three labs used the LIAISON^®^ 25 OH VITAMIN D TOTAL chemiluminescent immunoassay of DiaSorin, Saluggia, Italy. According to the manufacturer, the intraassay and inter-assay coefficients of variation were 5.4% and 10.6%, respectively, and the detection range was 10–375 nmol/L. Regarding comparable results across laboratories, we can state that all laboratories collaborating with the VICTORIA study took part in the quality assurance of the laboratory medical examinations of the Federal Medical Association (“Bundesärztekammer”) and conducted regular ring tests.

### 2.6. Statistical Analysis

The intention-to-treat (ITT) analysis included all randomized patients who provided a blood sample either on trial days 12–21 or trial weeks 13–16. The per-protocol (PP) analysis excluded study participants who failed to comply with the trial medication (<80% of capsules), who were falsely included, who discontinued treatment after visit 1 due to safety concerns prespecified in the protocols, were nonadherent, and who were taking any vitamin D product in addition to the trial medication. Due to the importance of protocol adherence for the relationship between the vitamin D supplementation and 25(OH)D level changes, the PP results are shown in the main paper and the ITT results in the Appendix A.

Assuming a normal distribution for the total serum 25(OH)D level, we performed a two-sample, two-tailed *t*-test for continuous outcomes, including the computation of means with a 95% CI. The *p*-value was derived by using the Satterthwaite method. If at least one event occurred during the trial in both trial arms, Fisher’s exact test was utilized for dichotomous outcomes to test for differences between the verum and placebo groups. A two-sided significance level of 0.04 was used for all tests in this interim analysis to leave a significance level of 0.01 for the main analysis when recruitment is completed. No multivariate models were used in the interim analysis.

## 3. Results

### 3.1. Participants’ Characteristics

This interim analysis included the first enrolled 74 study participants who completed the VICTORIA trial until 10 April 2022. Overall, 36 participants were randomly allocated to the placebo arm and 38 to the vitamin D_3_ arm. Due to missing blood samples, only 68 and 52 study participants could be included in the ITT analysis for laboratory measurement-based outcomes assessed at visits 1 and 2, respectively (Figure 1). Due to further exclusions, these numbers dropped to *n* = 64 and *n* = 41 for the PP analysis for visits 1 and 2, respectively.

Table 1 shows the baseline characteristics of the included participants. The mean age of all randomized participants was 61.8 years; 32.4% were female; and CRC stages I–III were approximately equally distributed (33.8% stage I, 29.7% stage II, and 29.7% stage III), whereas, for 6.8% of the patients, the cancer stage was not determined. However, it was known that these study participants were free of metastases, which was an exclusion criterion. With 28.2 kg/m^2^, the mean BMI was within the WHO’s definition of overweight (25- < 30 kg/m^2^). With 27.3 nmol/L, the mean 25(OH)D level was below the Institute of Medicines’ (IOM) deficiency threshold of 30 nmol/L. With 87.8 mL/min/1.73 m^2^, the mean eGFR was close to the optimal levels > 90 mL/min/1.73 m^2^. The mean albumin-corrected serum calcium and urinary calcium-to-creatinine ratio were far below the cut-offs for hypercalcemia and hypercalciuria stated in the Methods section. Of note, sufficient vitamin D status (25(OH)D levels ≥ 50 nmol/L) and severe renal dysfunction (eGRF < 30 mL/min/1.73 m^2^), as well as hypercalcemia and hypercalciuria, were exclusion criteria.

No relevant differences between the vitamin D_3_ and placebo arms were observed with respect to age, BMI, and laboratory-based factors. By chance, more females were included in the vitamin D (36.8%) than in the placebo arm (27.8%). Furthermore, the CRC stage distribution differed, with patients with stage II cancers being the largest group in the vitamin D arm (44.7%) and patients with stage I cancers being the largest group in the placebo arm (47.2%).

The median calculated loading dose for all analyzed trial participants (regardless of whether vitamin D_3_ or placebo was given) was 200,000 IU vitamin D_3_ (Interquartile range: 160,000–240,000), with large individual variations from 80,000 to 420,000 IU (Figure 2, all values were rounded up to the next 20,000 IU unit). For an illustration of the personalization of the loading dose, we stated the details of the ends of the distribution: The person who received a loading dose of 80,000 IU had a baseline 25(OH)D level of 48.7 nmol/L and a BMI of 24.5 kg/m^2^. In contrast, the person who received a loading dose of 420,000 IU had a baseline 25(OH)D level of 4.8 nmol/L and a BMI of 46.5 kg/m^2^.

### 3.2. Efficacy Endpoints

#### 3.2.1. Serum 25(OH)D Levels in the Total Trial Population

In the placebo group, the mean 25(OH)D levels (95% confidence interval (CI)) at screening (27.6 (23.6–31.6) nmol/L) did not change much until visit 1 (31.0 (27.2–34.7) nmol/L) and visit 2 (34.1 (27.1–41.1) nmol/L) (Figure 3 and Appendix A). In the verum group, the mean 25(OH)D levels (95% CI) at screening (25.9 (22.5–29.3) nmol/L) more than doubled until visit 1 (63.1 (58.1–68.0) nmol/L) and increased further until visit 2 (75.5 (69.2–81.9) nmol/L). The statistical tests for the 25(OH)D level comparisons between the two study arms were statistically significant at visits 1 and 2 (both *p* < 0.001) but not at the baseline (*p* = 0.501).

For subjects with repeated blood samples, the mean differences (95%CI) in the 25(OH)D levels from screening to visit 1 (3.3 (1.4–5.2) nmol/L) and from screening to visit 2 (5.5 (−2.1–13.1) nmol/L) were small in the placebo group (Figure 4 and Appendix A). The mean differences (95%CI) in the vitamin D_3_ group, however, were large, with an increase by 37.2 (31.8–42.5) nmol/L until visit 1 and by 45.0 (36.2–53.8) nmol/L until visit 2. The tests for comparisons between the two study arms were statistically significant at visits 1 and 2 (both *p* < 0.001).

The prevalence of vitamin D insufficiency (25(OH)D levels < 50 nmol/L) in the placebo group remained high at visit 1 (96.6%) and visit 2 (82.6%) (Table 2 and Appendix A). In contrast, only 20.0% of the study participants of the vitamin D_3_ group remained at the 25(OH)D levels < 50 nmol/L, and all of them had a sufficient vitamin D status at visit 2. The prevalence differences were highly statistically significant at both visits 1 and 2 (both *p* < 0.001).

#### 3.2.2. Serum 25(OH)D Levels in Patients with Vitamin D Deficiency at Enrolment

The box plots of the 25(OH)D levels during the course of the trial, restricted to subjects with vitamin D deficiency at screening (i.e., 25(OH)D < 30 nmol/L), are shown in Appendix A. The box plots at visits 1 and 2 were comparable to those obtained from the total population, but the mean changes (95%CI) in the 25(OH)D levels from screening to visit 1 (41.2 (35.1–47.3) nmol/L) and from screening to visit 2 (55.4 (41.5–69.2) nmol/L) were higher than among the total trial population with vitamin D insufficiency (i.e., 25(OH)D < 50 nmol/L).

### 3.3. Safety Endpoints

A tabulation of the safety events is shown in Table 3. No cases of hypervitaminosis D, hypercalcemia, or renal dysfunction were observed. Numerically, more cases of hypercalciuria were observed in the vitamin D_3_ than in the placebo group, but the difference was not statistically significant (*p* = 0.209). As specified in the protocols, treatment was discontinued for six patients with hypercalciuria after visit 1, and four of them provided repeated their blood and urine samples at visit 2, which showed a reduction of the urinary calcium-to-creatinine ratio to the levels at screening or even below (Appendix A). Importantly, the albumin-corrected serum calcium was not similarly increased in these six patients at visit 1, and the eGFR remained stable at high levels > 80 mL/min/1.73 m^2^ throughout the study (Appendix A).

Figure 5 shows the distribution of the urinary calcium-to-creatinine ratio during the course of the trial. While no change was observed in the placebo group from screening to visit 1, the mean and standard deviation increased for the intervention group. However, the urinary calcium-to-creatinine ratio decreased from screening to visit 2 for both the vitamin D_3_ and placebo arm, and the mean ratio was comparable for the two groups at visit 2. This might be partly explained by the treatment discontinuation of the six patients with hypercalciuria at visit 1. The mean urinary calcium-to-creatinine ratio difference between vitamin D_3_ and placebo group at visits 1 and 2 were not statistically significant (*p* = 0.152 and *p* = 0.618, respectively).

The means and distributions of the albumin-corrected serum calcium levels were very similar in the vitamin D_3_ and placebo groups during all study visits (Figure 6), and none of the statistical tests indicated a difference between the two groups at any time point (Appendix A).

In addition, the means and distributions of the eGFR were very similar in the two groups (Figure 7), and no statistically significant differences were observed at any time point (Appendix A).

## 4. Discussion

The combination of a personalized vitamin D_3_ loading dose, calculated with the equation of Jansen et al. [12], and a maintenance dose of 2000 IU for 12 weeks successfully treated vitamin D insufficiency in all CRC patients included in this interim analysis of the VICTORIA trial. The personalized loading dose elevated the 25(OH)D level’s substantially by, on average, 37 nmol/L during the first 11 days of the study, and 80% of patients already reached sufficient 25(OH)D levels ≥ 50 nmol/L at this early time point in the trial. All study participants reached sufficient vitamin D status after using the maintenance dose for 12 weeks. Among the safety parameters, only hypercalciuria occurred more frequently (not statistically significant) in the vitamin D_3_ group, but importantly, kidney function (needed to excrete high serum calcium) and the serum calcium levels were not affected by the intervention. The maximum 25(OH)D level observed in the trial population was 101 nmol/L, which is far from potentially harmful 25(OH)D levels > 150 nmol/L [2].

Regarding safety issues, previous clinical trials that administered very high bolus doses observed no single case of a clinically manifested overdose [12,19,20]. Overdoses have only been described in the literature for much higher cumulative vitamin D doses, typically between 2,220,000 and 6,360,000 IU [21]. The vitamin D intoxication dose reported in the SmPC of Dekristol^®^ 20,000 IU is stated to range between 40,000 and 100,000 IU per day administered over 1 to 2 months, which would be a cumulative dose between 2,440,000 and 6,100,000 IU and is consistent with the cited reports from the scientific literature. Thus, even more than the five-fold of the maximum applied vitamin D_3_ loading dose in our trial (420,000 IU) could still be considered safe if applied to patients without contraindications for vitamin D use.

We assume that the increase in urinary calcium levels after the absorption of the vitamin D_3_ loading dose is a normal physiological response of the body to the high initial dose, which will disappear within a few days to weeks after the dose is reduced to 2000 IU per day. Since the blood calcium levels did not rise at the same time, short-term hypercalciuria is not a recognizable safety risk, as long as the kidney function is not impaired and the kidney can eliminate the excess calcium. When we reported the first six cases of hypercalciuria to the competent authority for drug safety (Bundesinstitut für Arzneimittel und Medizinprodukte, Bonn, Germany), we were granted an amendment of the study protocols with respect to the treatment discontinuation rules for hypercalciuria. We now only discontinue treatment after visit 1 for individuals with hypercalciuria if, additionally, either the eGFR is below 60 mL/min/1.73 m^2^ or renal function has significantly deteriorated (decrease in eGFR ≥ 20% compared to eGFR at screening). None of the six cases from the interim analysis dataset would have needed to discontinue treatment according to these new rules. It will be of interest to see during the progress of the study if our assumption holds that the urinary calcium excretion of subjects with hypercalciuria after loading dose consumption normalizes within a few weeks, despite the intake of the maintenance dose of 2000 IU vitamin D_3_.

Regarding efficacy, the mean achieved 25(OH)D level by the personalized loading dose was lower in our trial (63.1 nmol/L) than the aspired 80 nmol/L, which was reached in the validation study of Jansen et al. for his equation 7 days after consumption of the loading dose (82 nmol/L). There are two potential explanations for this. First, since the blood samples for the measurement of the 25(OH)D levels were mostly taken in the morning of day 12 and sometimes in the morning of day 13, patients who took their last loading dose capsule on day 11 may not have fully absorbed and metabolized it. Although a vitamin D_3_ bolus is being absorbed quickly and most of the 25(OH)D increase is measurable in blood samples taken on the next day, approx. 18% of the total increase in 25(OH)D levels cannot be seen one day later and approx. 8% even not two days later, because the 25(OH)D levels peak on the third day after intake of the supplement [22]. In our trial, study participants who took vitamin D_3_ loading dose capsules until day 11 had, on average, 2.6 nmol/L lower 25(OH)D levels at visit 2 than patients who took loading dose capsules up to day 10. Thus, the timing of the blood sampling played a minor role. Second, differences in study populations may be more relevant. The study population of Jansen et at al. was recruited at the endocrinological outpatient clinic at Bispebjerg Hospital, University of Copenhagen, Denmark [12]. The patients attended the clinic because of vitamin D insufficiency or endocrinological diseases (primarily diabetes mellitus). There might be special patient characteristics among CRC patients that require higher vitamin D_3_ loading doses.

However, taken together with the maintenance dose of 2000 IU per day for 12 weeks, the applied loading dose equation of Jansen et al. [12] was perfectly suitable for CRC patients, because all of included CRC patients in our study reached sufficient 25(OH)D levels > 50 nmol/L in the end if they were randomized to the vitamin D_3_ group. It should be mentioned that there is no consensus among medical societies on the cut-off for sufficient 25(OH)D levels and, e.g., the Endocrine Society suggests using 75 nmol/L instead of the 50 nmol/L suggested by the IOM to define the sufficient 25(OH)D level [23]. This higher cut-off value would have led to a lower success rate in our study. However, as the association of 25(OH)D levels with adverse health outcomes, such as all-cause mortality, is not linear, and the excess risk is much higher in subjects with 25(OH)D levels < 50 nmol/L (especially in those with 25(OH)D < 30 nmol/L) than among subjects with 50- < 75 nmol/L, treating subjects with 25(OH)D levels < 50 nmol/L is of higher clinical relevance [24,25]. However, once the decision for a long-term treatment with vitamin D supplements was made, nothing speaks against aiming for 25(OH)D levels > 75 nmol/L.

We are aware of only one alternative equation for the personalization of a vitamin D_3_ loading dose, which has some similarities with the one of Jansen et al. [12]. Van Groningen et al. derived an equation based on the baseline 25(OH)D and body weight to target a 25(OH)D level of 75 nmol/L [10]:Loading dose = 40 × (target 25(OH)D level–baseline 25(OH)D level [nmol/L]) × body weight

The equation was derived from the general population without cancer and suboptimal vitamin D status [10]. A small validation study with nursing home inhabitants with vitamin D insufficiencies (25(OH)D < 50 nmol/L) applied a modified version of the equation of van Groningen by inserting 100 nmol/L instead of 75 nmol/L into the equation. Overall, 11 out of 14 (79%) study participants reached 25(OH)D levels > 75 nmol/L after 5 weeks [26]. Interestingly, although high personalized loading doses were applied (rounded median of calculated doses: 236,000 IU (IQR 185,000–251,000)), such as in our study, no changes in the albumin-corrected serum calcium and no differences in monitored adverse events rates compared with the control group were observed, and the maximum reached 25(OH)D levels were not much above 100 nmol/L. However, the urinary calcium levels were not assessed.

For a fair comparison of the equations of Jansen et al. and van Groningen et al., we applied the equation of van Groningen et al. to our study population and used the target 25(OH)D level of Jansen et al.’s equation (80 nmol/L). Thus, the following version of the equation of van Groningen et al. was used:Loading dose = 40 × (80–baseline 25(OH)D level [nmol/L]) × body weight

The comparison of the distribution of the personalized loading dose of the two equations in the total study population and stratified by obesity and vitamin D deficiency is shown in Appendix A. In the total trial population, the equation of van Groningen et al. would have yielded a 13% lower median loading dose than the equation of Jansen et al., and also, subjects requiring either low or high loading doses would have received less vitamin D_3_ if we had used the van Groningen equation. The gap between the two equations was similar for subjects with and without obesity. However, the von Groningen equation leads especially to lower loading doses than the Jansen equation for subjects with vitamin D deficiency, whereas the results were closer together for subjects with 25(OH)D levels between 30 and 50 nmol/L. Taking into consideration the importance of quickly raising the 25(OH)D levels of subjects with vitamin D deficiencies, we would not recommend the van Groningen equation for CRC patients.

Typically, vitamin D_3_ loading doses are administered as a large bolus at once, followed by a much lower maintenance dose. Such non-physiological high doses can lead to an upregulation of countervailing factors, which can ultimately lead to a lower synthesis or higher degradation of the biologically active hormone 1,25-dihdroxyvitamin D [27]. A recent systematic review and meta-analysis observed that vitamin D supplementation did not reduce cancer mortality in studies using large, intermittently administered bolus doses but in studies that used daily vitamin D dosing regimens [28]. We assume that two aspects are important for a vitamin D_3_ dosing regimen: a daily dosing regimen and the avoidance of non-physiological high doses. First, it should be specified that high initial bolus doses as loading doses are fine as long as they are administered daily and not with long pauses without treatment in between [29]. Second, we need to define the term “physiological dose”. We used the vitamin D_3_ equivalent to the amount of vitamin D_3_ the human body can naturally produce in the skin by sunbathing in a swimsuit for a whole day in the summer, which is 20,000 IU [30]. Thus, we would recommend consuming a loading dose with one capsule of 20,000 IU per day (e.g., 200,000 IU over 10 days and 400,000 IU over 20 days). Up to 3 weeks for most patients is still a relatively short time to overcome vitamin D insufficiency. The only reason why we allowed 40,000 IU per day in our trial was the fact that we could only insert 11 days for the consumption of the total loading dose in the 3-week rehabilitation clinic stay, because visit 1 needed to take place at the end of the rehabilitation, because blood and urine samples needed to be taken to check the safety parameters.

The maintenance dose of 2000 IU proved to be ideal for all patients in our 12-week trial. We could not conclude from our data that this dosage is also optimal in the long run, and we would rather assume that it should be personalized to the BMI of the patients as well. Physicians who would like to use our treatment regimen during their clinical routine can measure the 25(OH)D serum status every 3 months and adapt the daily vitamin D_3_ dose until a stable 25(OH)D level is reached in the target range of the IOM, which is from 50 to 150 nmol/L. The Endocrine Society Clinical Practice Guideline considers a 25(OH)D level ≥ 75 nmol/L as optimal [15]. It should additionally be mentioned that vitamin D_2_ could also potentially be used instead of vitamin D_3_, because they were equally effective in increasing the serum 25(OH)D levels in healthy adults aged 18–84 years at doses of 1000 IU daily [31]. However, our study used vitamin D_3_ in higher doses in diseased patients (CRC patients), and it cannot be taken for granted that vitamin D_2_ would have been as effective as vitamin D_3_ in our study.

The major strength of our study was the placebo-controlled randomized design allowing direct comparisons to an untreated group, which was also not allowed to co-supplement vitamin D using over-the-counter (OTC) preparations. The small 25(OH)D level increase in the placebo group could rather be explained by more sun exposure after rehabilitation due to an improved health status and a more active lifestyle than by undisclosed OTC vitamin D_3_ use, which would likely have manifested in more pronounced changes. The main limitation of this interim analysis of the VICTORIA trial was the low sample size, which led to less precise effect estimates as desired and not enough statistical power to detect a statistically significant difference for hypercalciuria in the two groups. Furthermore, other adverse events than those shown were recorded but will not be evaluated before recruitment of the total trial population is completed.

## 5. Conclusions

The applied personalized vitamin D_3_ loading dose, followed by a maintenance dose of 2000 IU, was safe and effectively treated vitamin D insufficiency in subjects with CRC.

## Figures and Tables

**Figure 1 nutrients-14-04546-f001:**
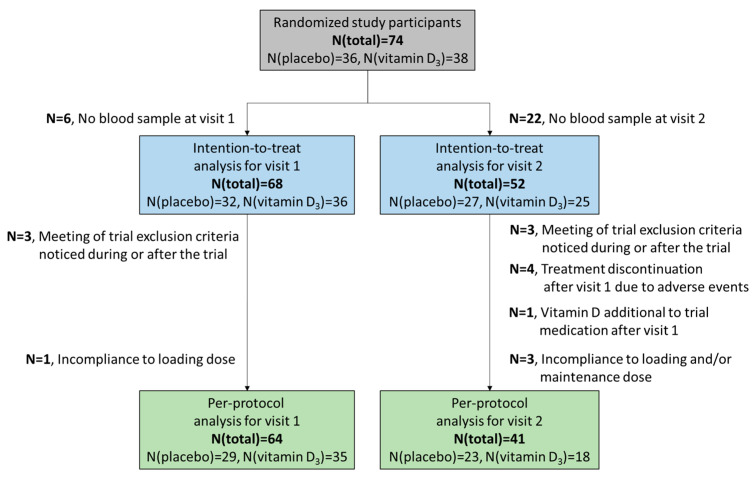
Flowchart of the study population.

**Figure 2 nutrients-14-04546-f002:**
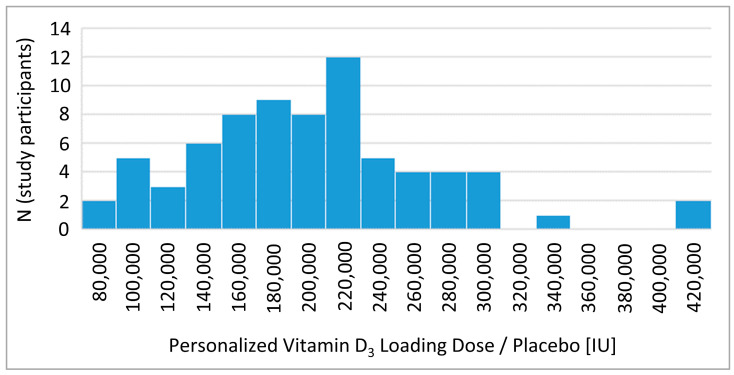
Distribution of the personalized loading dose. Note: Doses were rounded up to the next 20,000 unit. The histogram shows the loading doses of all randomized study participants, except one. The single study participant not shown is an outlier, because he/she was falsely included in the study (no vitamin D insufficiency at screening). Due to the study participants’ high 25(OH)D level of 61 nmol/L at screening, he/she received a vitamin D_3_/placebo loading dose of 40,000 IU.

**Figure 3 nutrients-14-04546-f003:**
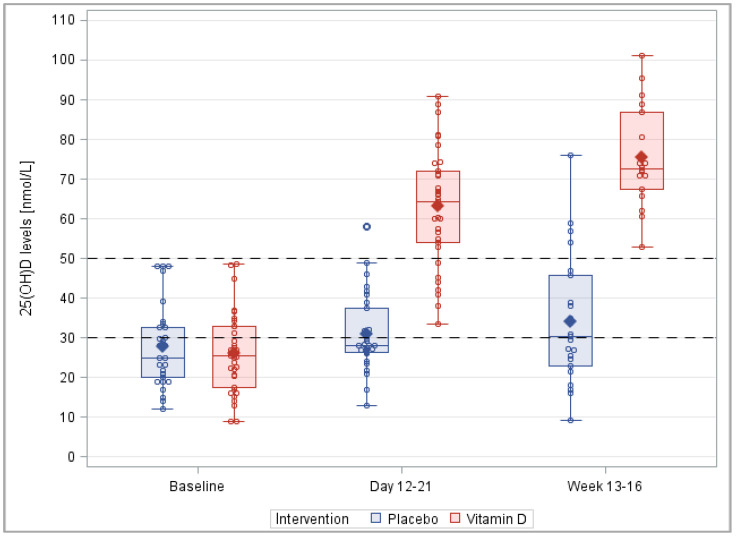
Box plots of the 25(OH)D levels during the course of the trial. Note: This figure is based on the detailed results of the PP analysis shown in Appendix A, which also shows the corresponding results of the ITT analysis.

**Figure 4 nutrients-14-04546-f004:**
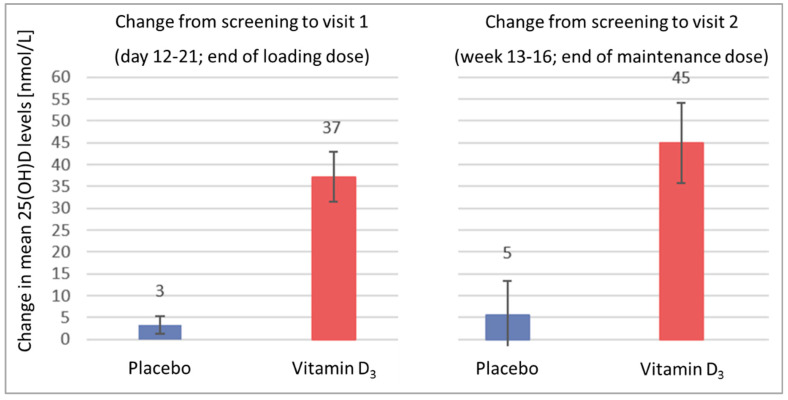
Change in the mean 25(OH)D levels with 95% confidence interval bars from screening to the end of rehabilitation (visit 1, end of loading dose, days 12–21) and from screening to end of the study (visit 2, end of maintenance dose, weeks 13–16). Note: This figure is based on the detailed results of the PP analysis shown in Appendix A, which also shows the corresponding results of the ITT analysis. A more detailed display of the changes in the 25(OH)D levels in the box plots is shown in Appendix A.

**Figure 5 nutrients-14-04546-f005:**
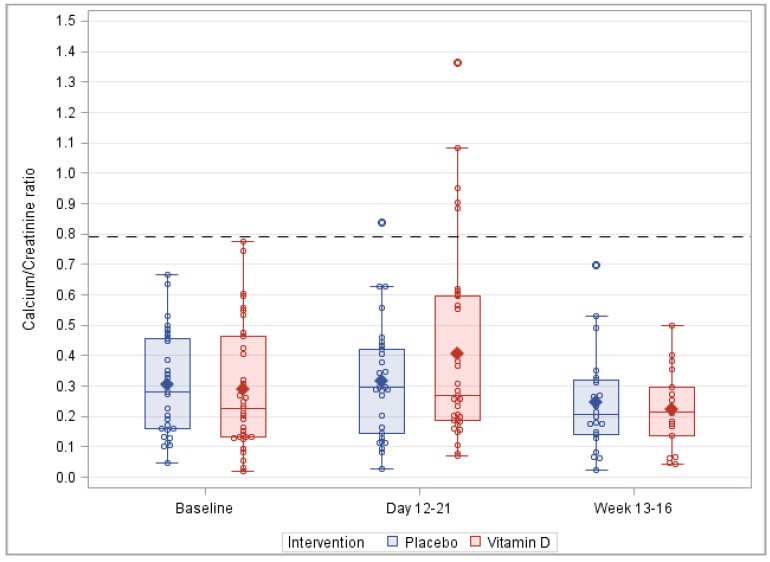
Box plots of the urinary calcium-to-creatinine ratio during the course of the trial. Note: This figure is based on the detailed results of the PP analysis shown in Appendix A, which also shows the corresponding results of the ITT analysis.

**Figure 6 nutrients-14-04546-f006:**
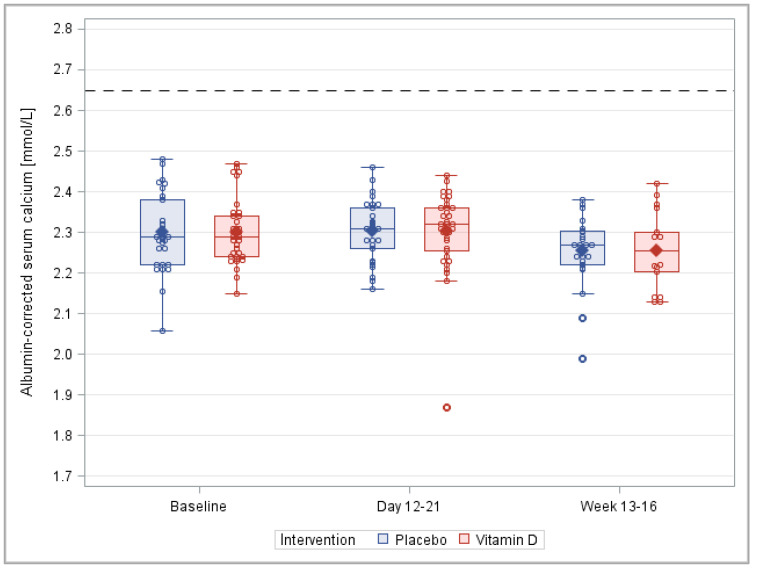
Box plots of the albumin-corrected serum calcium during the course of the trial. Note: This figure is based on the detailed results of the PP analysis shown in Appendix A, which also shows the corresponding results of the ITT analysis and the statistical test results.

**Figure 7 nutrients-14-04546-f007:**
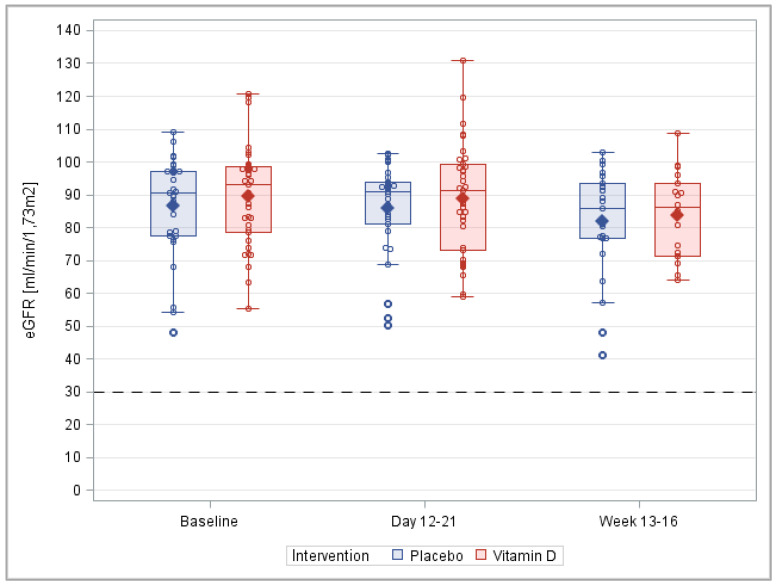
Box plots of the eGFR during the course of the trial. Note: This figure is based on the detailed results of the PP analysis shown in Appendix A, which also shows the corresponding results of the ITT analysis and the statistical test results.

**Table 1 nutrients-14-04546-t001:** Baseline characteristics of participants of the interim analysis of the VICTORIA trial.

	All Randomized (N = 74)	Vitamin D_3_ (N = 38)	Placebo (N = 36)
Age [years], mean (SD)	61.8 (9.8)	62.1 (9.9)	61.6 (9.7)
Sex			
Female, *n*, %	24 (32.4)	14 (36.8)	10 (27.8)
Male, *n*, %	50 (67.6)	24 (63.2)	26 (72.2)
Cancer stage			
I, *n*, %	25 (33.8)	8 (21.1)	17 (47.2)
II, *n*, %	22 (29.7)	17 (44.7)	5 (13.9)
III, *n*, %	22 (29.7)	11 (28.9)	11 (30.6)
Unknown, *n*, %	5 (6.8)	2 (5.3)	3 (8.3)
BMI (kg/m^2^), mean (SD)	28.2 (5.8)	28.5 (6.1)	28.0 (5.5)
25(OH)D level (nmol/L), mean (SD)	27.3 (10.7)	26.5 (9.9)	28.2 (11.6)
Albumin-corrected serum calcium (mmol/L), mean (SD)	2.29 (0.10)	2.29 (0.09)	2.29 (0.11)
Urinary calcium-to-creatinine ratio (mmol/mmol), mean (SD)	0.28 (0.19)	0.28 (0.20)	0.28 (0.17)
Estimated glomerular filtration rate (ml/min/1.73 m^2^), mean (SD)	87.8 (15.0)	89.0 (15.4)	86.5 (14.6)

Abbreviations: SD, standard deviation.

**Table 2 nutrients-14-04546-t002:** Prevalence of vitamin insufficiency during the course of the trial.

Trial Arm	N	Vitamin D Insufficiency		*p* ^a^
		(25(OH)D < 50 nmoL/L)		
		No N (%)	Yes N (%)	
**Visit 1 (Days 12–21; end of loading dose)**				
Placebo	29	1 (3.5)	28 (96.6)	<0.001
Vitamin D_3_	35	28 (80.0)	7 (20.0)	
**Visit 2 (Weeks 13–16; end of maintenance dose)**				
Placebo	23	4 (17.4)	19 (82.6)	<0.001
Vitamin D_3_	18	18 (100.0)	0 (0.0)	

Abbreviations: 25(OH)D, 25-hydroxyvitamin D; NA, not applicable. ^a^ Fisher’s exact test. Statistically significant in the interim analysis if *p* < 0.04. Note: This table is based on the detailed results of the PP analysis shown in Appendix A, which also shows the corresponding results of the ITT analysis.

**Table 3 nutrients-14-04546-t003:** Safety events during the course of the trial.

Trial Arm	N	Hypervitaminosis D ^1^	Hypercalcemia ^2^	Hypercalciuria ^3^	Renal Dysfunction ^4^
		N (%)	N (%)	N (%)	N (%)
**Visit 1; end of loading dose; days 12–21**					
Placebo	29	0 (0)	0 (0)	1 (3.4)	0 (0)
Vitamin D_3_	35	0 (0)	0 (0)	5 (14.3)	0 (0)
**Visit 2; end of maintenance dose; weeks 13–16**					
Placebo	23	0 (0)	0 (0)	0 (0)	0 (0)
Vitamin D_3_	18	0 (0)	0 (0)	0 (0)	0 (0)

^1^ 25(OH)D levels > 150 nmol/L [2]. ^2^ Albumin-corrected serum calcium > 2.65 mmol/L [16]. ^3^ Random urine calcium ≥ 0.79 mmol/mmol creatinine [17]. ^4^ eGFR < 30 mL/min/1.73 m^2^ [18]. Note: This table shows the results of the PP analysis. The ITT analysis had the same number of adverse events. Due to different sample sizes in the ITT analysis, the prevalence of hypercalciuria at visit 1 was 13.9% in the vitamin D3 group vs. 3.1% in the placebo group (*p* = 0.203).

## Data Availability

The data will not be published on an open-access platform. After completion of the study, interested scientists can request data use and receive pseudonymized data upon approval of this application by the sponsor.

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
