# Peer review of "Efficacy and Safety of a Personalized Vitamin D3 Loading Dose Followed by Daily 2000 IU in Colorectal Cancer Patients with Vitamin D Insufficiency: Interim Analysis of a Randomized Controlled Trial"

_nutrients, 2022, doi:10.3390/nu14214546_

Round 1

Reviewer 1 Report

This manuscript reports increases in 25(OH)D concentrations in cancer patients using a loading dose followed by a maintenance dose. The experiment was conducted well and the increases in 25(OH)D were as expected. The findings will be useful in treating cancer patients as well as, perhaps, patients with other diseases.

With 27.3 nmol/L, the mean 25(OH)D level was below the Institute of 174
Medicines’ (IOM) deficiency threshold of 30 nmol/L.

Comment: I cannot find that number in the IOM journal publications by Ross et al.

IOM Committee Members Respond to Endocrine Society ...

https://academic.oup.com › jcem › article

by CJ Rosen · 2012 · Cited by 647 — The guideline specifies that vitamin D deficiency is present when serum levels of 25OHD are below 20 ng/ml. This conclusion is stated as a fact for the general ...

Disagreement: should target... · ‎Disagreement: how should... · ‎Summary

What assay was used for 25(OH)D measurements and what was its accuracy, etc.

As vitamin D insufficiency is usually neither diagnosed nor treated in CRC pa-

53

tients, there is now evidence that it should be treated but the optimal dosing regimen is

54

unknown.

Suggest looking at these papers

Association of 25-hydroxyvitamin D level with survival outcomes in female breast cancer patients: A meta-analysis.

Li C, Li H, Zhong H, Li X.J Steroid Biochem Mol Biol. 2021 Sep;212:105947. doi: 10.1016/j.jsbmb.2021.105947. 

Sufficient 25-Hydroxyvitamin D Levels 2 Years after Colorectal Cancer Diagnosis are Associated with a Lower Risk of All-cause Mortality.

Wesselink E, Kok DE, de Wilt JHW, Bours MJL, van Zutphen M, Keulen ETP, Kruyt FM, Breukink SO, Kouwenhoven EA, van den Ouweland J, Weijenberg MP, Kampman E, van Duijnhoven FJB.Cancer Epidemiol Biomarkers Prev. 2021 Apr;30(4):765-773. doi: 10.1158/1055-9965.EPI-20-1388. 

Associations of vitamin D status with colorectal cancer risk and survival.

Zhou J, Ge X, Fan X, Wang J, Miao L, Hang D.Int J Cancer. 2021 Aug 1;149(3):606-614. doi: 10.1002/ijc.33580. 

Si-Min Peng 1Na Yu 2Jun Che 3Jia-Ying Xu 4Guo-Chong Chen 1Da-Peng Li 5Yu-Song Zhang 6Li-Qiang Qin 7Total, bioavailable and free 25-hydroxyvitamin D are associated with the prognosis of patients with non-small cell lung cancer. Cancer Causes Control2022 Jul;33(7):983-993.  doi: 10.1007/s10552-022-01579-6. Epub 2022 Apr 11.

Stephanie J Weinstein 1Alison M Mondul 2Tracy M Layne 3Kai Yu 1Jiaqi Huang 1Rachael Z Stolzenberg-Solomon 1Regina G Ziegler 1Mark P Purdue 1Wen-Yi Huang 1Christian C Abnet 1Neal D Freedman 1Demetrius Albanes 1Prediagnostic Serum Vitamin D, Vitamin D Binding Protein Isoforms, and Cancer Survival JNCI Cancer Spectr2022 Mar 2;6(2):pkac019.  doi: 10.1093/jncics/pkac019.

'Palliative-D'-Vitamin D Supplementation to Palliative Cancer Patients: A Double Blind, Randomized Placebo-Controlled Multicenter Trial.

Helde Frankling M, Klasson C, Sandberg C, Nordström M, Warnqvist A, Bergqvist J, Bergman P, Björkhem-Bergman L.Cancers (Basel). 2021 Jul 23;13(15):3707. doi: 10.3390/cancers13153707.

De novo vitamin D supplement use post-diagnosis is associated with breast cancer survival.

Madden JM, Murphy L, Zgaga L, Bennett K.Breast Cancer Res Treat. 2018 Nov;172(1):179-190. doi: 10.1007/s10549-018-4896-6.

In the U.S., physicians are more likely to prescribe vitamin D2 to cancer patients than vitamin D3 since it is readily available in pharmacies, even though it costs more. Would it be useful to briefly discuss that vitamin D3 is better than vitamin D2 in this case?

Regarding daily loading dose, Keum et al. did not consider bolus dose followed by maintenance doses. It might be worthwhile to look at other articles such as:

Rapid and Effective Vitamin D Supplementation May Present Better Clinical Outcomes in COVID-19 (SARS-CoV-2) Patients by Altering Serum INOS1, IL1B, IFNg, Cathelicidin-LL37, and ICAM1.

Gönen MS, AlaylıoÄŸlu M, Durcan E, Özdemir Y, Åžahin S, KonukoÄŸlu D, Nohut OK, Ürkmez S, Küçükece B, Balkan Ä°Ä°, Kara HV, Börekçi Åž, Özkaya H, Kutlubay Z, Dikmen Y, Keskindemirci Y, Karras SN, Annweiler C, Gezen-Ak D, Dursun E.Nutrients. 2021 Nov 12;13(11):4047. doi: 10.3390/nu13114047.

Vitamin D: A single initial dose is not bogus if followed by an appropriate maintenance intake.

Wimalawansa SJ, Whittle R.JBMR Plus. 2022 Feb 18;6(3):e10606. doi: 10.1002/jbm4.10606. eCollection 2022 Mar.

Duplicate reference; 2015 appears to be the correct one.

10. Heaney, R.P.; Armas, L.A. Quantifying the vitamin D economy. Nutr. Rev. 2015, 73, 51-67, doi:10.1093/nutrit/nuu004. 502

 13. Heaney, R.P.; Armas, L.A.G. Quantifying the vitamin D economy. Nutr. Rev. 2014, 73, 51-67, doi:10.1093/nutrit/nuu004. 509

Significant digits. The general rule is that no more non-zero digits should be given than are justified by the uncertainty of the value.

See "Too many digits: the presentation of numerical data"

https://www.ncbi.nlm.nih.gov/pmc/articles/PMC4483789/

If the uncertainty is greater than about 7%, only two non-zero digits are justified.

P values should be given to two decimal places unless the first two are 00 or the number lies between 0.045 and 0.054.

Thus, in Table 1, with less than 150 participants, percentages should be given in whole numbers.

25(OH)D level [nmol/L], mean (SD)

27.3 (10.7)

26.5 (9.9)

28.2 (11.6)

Should be

25(OH)D level [nmol/L], mean (SD)

27 (11)

27 (10)

28 (12)

Interestingly, although high personalized loading doses were applied (median:

376

235,754 IU (IQR 184,534–251,194)),

Should be

Interestingly, although high personalized loading doses were applied (median:

376

 240,000 IU (IQR 190,000–250,000)),

Please review all numbers in abstract, text, tables, and figures and adjust accordingly.

Author Response

General appraisal

This manuscript reports increases in 25(OH)D concentrations in cancer patients using a loading dose followed by a maintenance dose. The experiment was conducted well and the increases in 25(OH)D were as expected. The findings will be useful in treating cancer patients as well as, perhaps, patients with other diseases.

Response

Dear reviewer,

Thank you very much for your review and your positive general appraisal of our article. We took utmost care to address all your comments and you can see the responses below. The changes in the manuscript are highlighted with yellow colour.

Point 1

With 27.3 nmol/L, the mean 25(OH)D level was below the Institute of 174
Medicines’ (IOM) deficiency threshold of 30 nmol/L.

Comment: I cannot find that number in the IOM journal publications by Ross et al.

IOM Committee Members Respond to Endocrine Society ...https://academic.oup.com › jcem › article by CJ Rosen · 2012 · Cited by 647 — The guideline specifies that vitamin D deficiency is present when serum levels of 25OHD are below 20 ng/ml. This conclusion is stated as a fact for the general ...

Response to point 1:

There must have been an error in the paper you cited regarding the definition of vitamin D deficiency. In the original guideline of the IOM (available for download here: https://nap.nationalacademies.org/catalog/13050/dietary-reference-intakes-for-calcium-and-vitamin-d), we cited, it says in the Summary, page 13: “This committee’s review of data suggests that persons are at risk of deficiency relative to bone health at serum 25OHD levels of below 30 nmol/L (12 ng/mL). Some, but not all, persons are potentially at risk for inadequacy at serum 25OHD levels between 30 and 50 nmol/L (12 and 20 ng/mL)”.

Point 2

What assay was used for 25(OH)D measurements and what was its accuracy, etc.

Response to point 2:

These details are now added to the manuscript (page 3, lines 143-153).

Point 3

“As vitamin D insufficiency is usually neither diagnosed nor treated in CRC patients, there is now evidence that it should be treated but the optimal dosing regimen is unknown.”  

Suggest looking at these papers

Association of 25-hydroxyvitamin D level with survival outcomes in female breast cancer patients: A meta-analysis.

Li C, Li H, Zhong H, Li X.J Steroid Biochem Mol Biol. 2021 Sep;212:105947. doi: 10.1016/j.jsbmb.2021.105947. 

Sufficient 25-Hydroxyvitamin D Levels 2 Years after Colorectal Cancer Diagnosis are Associated with a Lower Risk of All-cause Mortality.

Wesselink E, Kok DE, de Wilt JHW, Bours MJL, van Zutphen M, Keulen ETP, Kruyt FM, Breukink SO, Kouwenhoven EA, van den Ouweland J, Weijenberg MP, Kampman E, van Duijnhoven FJB.Cancer Epidemiol Biomarkers Prev. 2021 Apr;30(4):765-773. doi: 10.1158/1055-9965.EPI-20-1388. 

Associations of vitamin D status with colorectal cancer risk and survival.

Zhou J, Ge X, Fan X, Wang J, Miao L, Hang D.Int J Cancer. 2021 Aug 1;149(3):606-614. doi: 10.1002/ijc.33580. 

Si-Min Peng 1, Na Yu 2, Jun Che 3, Jia-Ying Xu 4, Guo-Chong Chen 1, Da-Peng Li 5, Yu-Song Zhang 6, Li-Qiang Qin 7Total, bioavailable and free 25-hydroxyvitamin D are associated with the prognosis of patients with non-small cell lung cancer. Cancer Causes Control. 2022 Jul;33(7):983-993.  doi: 10.1007/s10552-022-01579-6. Epub 2022 Apr 11.

Stephanie J Weinstein 1, Alison M Mondul 2, Tracy M Layne 3, Kai Yu 1, Jiaqi Huang 1, Rachael Z Stolzenberg-Solomon 1, Regina G Ziegler 1, Mark P Purdue 1, Wen-Yi Huang 1, Christian C Abnet 1, Neal D Freedman 1, Demetrius Albanes 1Prediagnostic Serum Vitamin D, Vitamin D Binding Protein Isoforms, and Cancer Survival JNCI Cancer Spectr. 2022 Mar 2;6(2):pkac019.  doi: 10.1093/jncics/pkac019.

'Palliative-D'-Vitamin D Supplementation to Palliative Cancer Patients: A Double Blind, Randomized Placebo-Controlled Multicenter Trial.

Helde Frankling M, Klasson C, Sandberg C, Nordström M, Warnqvist A, Bergqvist J, Bergman P, Björkhem-Bergman L.Cancers (Basel). 2021 Jul 23;13(15):3707. doi: 10.3390/cancers13153707.

De novo vitamin D supplement use post-diagnosis is associated with breast cancer survival.

Madden JM, Murphy L, Zgaga L, Bennett K.Breast Cancer Res Treat. 2018 Nov;172(1):179-190. doi: 10.1007/s10549-018-4896-6.

Response to point 3:

Thank you for pointing us towards this interesting literature. However, in this sentence, we wanted to cite only studies about the relationship of vitamin D and the prognosis of colorectal cancer. Thus, we only added citations of the studies of Wesselink et al. and Zhou et al. to the introduction (lines 48-50 and line 57). 

Point 4 

In the U.S., physicians are more likely to prescribe vitamin D2 to cancer patients than vitamin D3 since it is readily available in pharmacies, even though it costs more. Would it be useful to briefly discuss that vitamin D3 is better than vitamin D2 in this case?

Response to point 4

Done as suggested (p. 14, lines 453-458).

Point 5

Regarding daily loading dose, Keum et al. did not consider bolus dose followed by maintenance doses. It might be worthwhile to look at other articles such as:

Rapid and Effective Vitamin D Supplementation May Present Better Clinical Outcomes in COVID-19 (SARS-CoV-2) Patients by Altering Serum INOS1, IL1B, IFNg, Cathelicidin-LL37, and ICAM1.

Gönen MS, AlaylıoÄŸlu M, Durcan E, Özdemir Y, Åžahin S, KonukoÄŸlu D, Nohut OK, Ürkmez S, Küçükece B, Balkan Ä°Ä°, Kara HV, Börekçi Åž, Özkaya H, Kutlubay Z, Dikmen Y, Keskindemirci Y, Karras SN, Annweiler C, Gezen-Ak D, Dursun E.Nutrients. 2021 Nov 12;13(11):4047. doi: 10.3390/nu13114047.

Vitamin D: A single initial dose is not bogus if followed by an appropriate maintenance intake.

Wimalawansa SJ, Whittle R.JBMR Plus. 2022 Feb 18;6(3):e10606. doi: 10.1002/jbm4.10606. eCollection 2022 Mar.

Response to point 5

Thank you for the suggested references. First, we would like to clarify that Keum et al. criticized “large, intermittently administered bolus doses”. The intermittently use aspect is important and we now used the suggested article of Wimalawansa et al. to clarify that bolos doses are fine as long as they are taken daily (page 13, lines 433-435). The other suggested study of Gönen et al. used a daily high-dose vitamin D3 dosing regimen for a short time (up to 14 days), which is not comparable with our study because they only used different vitamin D3 doses depending on the initial 25(OH)D level but did not consider the study participant’s weight. In addition, the study was conducted in very specific population (COVID-19 patients in an acute care scenario), which is not generalizable to other populations.   

Point 6

Duplicate reference; 2015 appears to be the correct one.

  1. Heaney, R.P.; Armas, L.A. Quantifying the vitamin D economy. Nutr. Rev. 2015, 73, 51-67, doi:10.1093/nutrit/nuu004. 502
  2. Heaney, R.P.; Armas, L.A.G. Quantifying the vitamin D economy. Nutr. Rev. 2014, 73, 51-67, doi:10.1093/nutrit/nuu004. 509

 Response to point 6

 Thank you for pointing to this mistake that we now corrected.

Point 7 

Significant digits. The general rule is that no more non-zero digits should be given than are justified by the uncertainty of the value.  See "Too many digits: the presentation of numerical data" https://www.ncbi.nlm.nih.gov/pmc/articles/PMC4483789/

If the uncertainty is greater than about 7%, only two non-zero digits are justified. P values should be given to two decimal places unless the first two are 00 or the number lies between 0.045 and 0.054.

Thus, in Table 1, with less than 150 participants, percentages should be given in whole numbers.

Line 376 “Interestingly, although high personalized loading doses were applied (median: 235,754 IU (IQR 184,534–251,194)),” should be “Interestingly, although high personalized loading doses were applied (median: 240,000 IU (IQR 190,000–250,000)),”.

Please review all numbers in abstract, text, tables, and figures and adjust accordingly.

Response to point 7

We reviewed all figures in the article and changed the rounded numbers where we found it appropriate. All p-values are now presented with no more than 3 digits after the comma (page 6, line 228; page 7, line 240; page 8, line 253; page 8, Table 2) and the loading dose mentioned previously in line 376 was rounded to full 1,000 IU (page 13, lines 403-404). We did not round percentages to full numbers because stating 1 digit after the comma is the conventional standard for percentages.

Reviewer 2 Report

It's an interesting and valuable paper and it puts some novelty to clinical practice. However, some corrections are suggested:

1. "Study design and participants" needs more information regarding the study population. I.e., could all patients with any histological type, any grade of differentiation of CRC, or any type of surgery (endoscopic or laparotomic) have been included in the trial? The paper referenced in [11], as a source describing the protocol in details, also doesn't give clear answer.

2. Table 1. It is not clear which parameters in the verum group were statistically significantly different from the placebo group. Please provide the appropriate p-values for each parameter.

3. Many authors suggest that for optimal health, 25-OH-vitamin D values should be at least 75 nmol/l or even at least 100 nmol/l, particularly for cancer patients [1-8]. Therefore, the selected 50 nmol/l value as cut-off of vitamin D sufficiency, is questionable. Please discuss this aspect in the "Discussion" chapter.

4. It should be mentioned in the "Introduction", that vitamin D is also useful for CRC patients in other ways, e.g. it might increase the efficacy of chemotherapy and/or reduce the adverse reactions of chemotherapy [9].

References:

1.         Ebadi, M. and A.J. Montano-Loza, Perspective: improving vitamin D status in the management of COVID-19. Eur J Clin Nutr, 2020. 74(6): p. 856-859.

2.         Grant, W.B., et al., Evidence that Vitamin D Supplementation Could Reduce Risk of Influenza and COVID-19 Infections and Deaths. Nutrients, 2020. 12(4): p. 988.

3.         Wimalawansa, S.J., Global epidemic of coronavirus—COVID-19: what can we do to minimize risks. Eur. J. Biomed. Pharm. Sci., 2020. 7(3): p. 432–438.

4.         Charoenngam, N. and M.F. Holick, Immunologic Effects of Vitamin D on Human Health and Disease. Nutrients, 2020. 12(7): p. 2097.

5.         Pludowski, P., et al., Vitamin D supplementation guidelines. J Steroid Biochem Mol Biol, 2018. 175: p. 125-135.

6.         Amrein, K., et al., Vitamin D deficiency 2.0: an update on the current status worldwide. Eur J Clin Nutr, 2020. 74(11): p. 1498-1513.

7.         Kimball, S.M. and M.F. Holick, Official recommendations for vitamin D through the life stages in developed countries. Eur J Clin Nutr, 2020. 74(11): p. 1514-1518.

8.         Grober, U. and M.F. Holick, The coronavirus disease (COVID-19) - A supportive approach with selected micronutrients. Int J Vitam Nutr Res, 2021: p. 1-22.

9.         Peng, J., et al., Effects of vitamin D on drugs: Response and disposal. Nutrition, 2020. 74: p. 110734.

Author Response

General appraisal

It's an interesting and valuable paper and it puts some novelty to clinical practice. However, some corrections are suggested:

Response

Dear reviewer,

Thank you very much for your review and your positive general appraisal of our article. We took utmost care to address all your comments and you can see the responses below. The changes in the manuscript are highlighted with yellow colour.

Point 1

"Study design and participants" needs more information regarding the study population. I.e., could all patients with any histological type, any grade of differentiation of CRC, or any type of surgery (endoscopic or laparotomic) have been included in the trial? The paper referenced in [11], as a source describing the protocol in details, also doesn't give clear answer.

Response to point 1

We now added the details asked for (page 2, lines 87-88: “Eligible patients have a diagnosis of non-metastatic colorectal cancer (not stage IV), a tumor surgery within the past year (type of surgery not specified), and vitamin D insufficiency (25(OH)D) levels < 50 nmol/L) at the time of screening.” ).

Point 2

Table 1. It is not clear which parameters in the verum group were statistically significantly different from the placebo group. Please provide the appropriate p-values for each parameter.

Response to point 2

Unfortunately, it is not possible to add the p-values to table 1 in this interim analysis of the RCT because we want to present p-values for potential differences in the baseline characteristics in the future paper with the whole trial population. This would be conduction of the same statistical tests twice and the alpha error of 0.05 can only be used once (which we want to do in the future main paper).

Point 3

Many authors suggest that for optimal health, 25-OH-vitamin D values should be at least 75 nmol/l or even at least 100 nmol/l, particularly for cancer patients [1-8]. Therefore, the selected 50 nmol/l value as cut-off of vitamin D sufficiency, is questionable. Please discuss this aspect in the "Discussion" chapter.

Response to point 3

A discussion of the cut-offs for the definition of sufficient 25(OH)D levels was added (page 12, lines 382-392). 

Point 4

It should be mentioned in the "Introduction", that vitamin D is also useful for CRC patients in other ways, e.g. it might increase the efficacy of chemotherapy and/or reduce the adverse reactions of chemotherapy [9].

Response to point 4

Thank you. This important aspect was added (page 2, lines 53-55).

References:

  1. Ebadi, M. and A.J. Montano-Loza, Perspective: improving vitamin D status in the management of COVID-19. Eur J Clin Nutr, 2020. 74(6): p. 856-859.
  2. Grant, W.B., et al., Evidence that Vitamin D Supplementation Could Reduce Risk of Influenza and COVID-19 Infections and Deaths. Nutrients, 2020. 12(4): p. 988.
  3. Wimalawansa, S.J., Global epidemic of coronavirus—COVID-19: what can we do to minimize risks. Eur. J. Biomed. Pharm. Sci., 2020. 7(3): p. 432–438.
  4. Charoenngam, N. and M.F. Holick, Immunologic Effects of Vitamin D on Human Health and Disease. Nutrients, 2020. 12(7): p. 2097.
  5. Pludowski, P., et al., Vitamin D supplementation guidelines. J Steroid Biochem Mol Biol, 2018. 175: p. 125-135.
  6. Amrein, K., et al., Vitamin D deficiency 2.0: an update on the current status worldwide. Eur J Clin Nutr, 2020. 74(11): p. 1498-1513.
  7. Kimball, S.M. and M.F. Holick, Official recommendations for vitamin D through the life stages in developed countries. Eur J Clin Nutr, 2020. 74(11): p. 1514-1518.
  8. Grober, U. and M.F. Holick, The coronavirus disease (COVID-19) - A supportive approach with selected micronutrients. Int J Vitam Nutr Res, 2021: p. 1-22.
  9. Peng, J., et al., Effects of vitamin D on drugs: Response and disposal. Nutrition, 2020. 74: p. 110734.